# Stilbenes, a Versatile Class of Natural Metabolites for Inflammation—An Overview

**DOI:** 10.3390/molecules28093786

**Published:** 2023-04-28

**Authors:** Jameel M. Al-Khayri, Roseanne Mascarenhas, Himanshu Madapur Harish, Yashwanth Gowda, Vasantha Veerappa Lakshmaiah, Praveen Nagella, Muneera Qassim Al-Mssallem, Fatima Mohammed Alessa, Mustafa Ibrahim Almaghasla, Adel Abdel-Sabour Rezk

**Affiliations:** 1Department of Agricultural Biotechnology, College of Agriculture and Food Sciences, King Faisal University, Al-Ahsa 31982, Saudi Arabia; arazk@kfu.edu.sa; 2Department of Life Sciences, CHRIST (Deemed to Be University), Bangalore 560029, India; roseanne.mascarenhas@mbty.christuniversity.in (R.M.); himanshu.mh@mzoo.christuniversity.in (H.M.H.); yashwanth.gowda@mzoo.christuniversity.in (Y.G.); vasantha.vl@christuniversity.in (V.V.L.); 3Department of Food Science and Nutrition, College of Agriculture and Food Sciences, King Faisal University, Al-Ahsa 31982, Saudi Arabia; mmssallem@kfu.edu.sa (M.Q.A.-M.); falissa@kfu.edu.sa (F.M.A.); 4Department of Arid Land Agriculture, College of Agriculture and Food Sciences, King Faisal University, Al-Ahsa 31982, Saudi Arabia; malmghaslah@kfu.edu.sa; 5Plant Pests, and Diseases Unit, College of Agriculture and Food Sciences, King Faisal University, Al-Ahsa 31982, Saudi Arabia; 6Department of Virus and Phytoplasma, Plant Pathology Institute, Agricultural Research Center, Giza 12619, Egypt

**Keywords:** stilbenes, inflammation, plant secondary metabolites, resveratrol, viniferins, *Vitis vinifera*

## Abstract

Stilbenes are polyphenolic allelochemicals synthesized by plants, especially grapes, peanuts, rhubarb, berries, etc., to defend themselves under stressful conditions. They are now exploited in medicine for their antioxidant, anti-proliferative and anti-inflammatory properties. Inflammation is the immune system’s response to invading bacteria, toxic chemicals or even nutrient-deprived conditions. It is characterized by the release of cytokines which can wreak havoc on healthy tissues, worsening the disease condition. Stilbenes modulate NF-κB, MAPK and JAK/STAT pathways, and reduce the transcription of inflammatory factors which result in maintenance of homeostatic conditions. Resveratrol, the most studied stilbene, lowers the Michaelis constant of SIRT1, and occupies the substrate binding pocket. Gigantol interferes with the complement system. Besides these, oxyresveratrol, pterostilbene, polydatin, viniferins, etc., are front runners as drug candidates due to their diverse effects from different functional groups that affect bioavailability and molecular interactions. However, they each have different thresholds for toxicity to various cells of the human body, and thus a careful review of their properties must be conducted. In animal models of autoinflammatory diseases, the mode of application of stilbenes is important to their absorption and curative effects, as seen with topical and microemulsion gel methods. This review covers the diversity seen among stilbenes in the plant kingdom and their mechanism of action on the different inflammatory pathways. In detail, macrophages’ contribution to inflamed conditions in the liver, the cardiac, connective and neural tissues, in the nephrons, intestine, lungs and in myriad other body cells is explored, along with detailed explanation on how stilbenes alleviate the symptoms specific to body site. A section on the bioavailability of stilbenes is included for understanding the limitations of the natural compounds as directly used drugs due to their rapid metabolism. Current delivery mechanisms include sulphonamides, or using specially designed synthetic drugs. It is hoped that further research may be fueled by this comprehensive work that makes a compelling argument for the exploitation of these compounds in medicine.

## 1. Introduction

Modern medicine has made great strides towards creating synthetic drugs that are quick and efficient, but more often than not they come with the catch of severe side effects. Thus, pharmaceutical studies are examining natural compounds, especially from plant sources, which have the potential to alleviate diseases with minimum negative effects once in the purified state. Plants synthesize polyphenolic substances called stilbenes to defend themselves from pathogens, bacterial and fungal growth [1] and to shield themselves against the damaging effects of UV light. Thus, they are phytoalexins. They are characterized by a carbon skeleton of 1,2-diphenylethylene (C6–C2–C6), consisting of an ethylene moiety in the middle of two benzene rings [2] (Figure 1). Resveratrol (RSV) is the most well-known of the stilbenes, especially because of its role in the French paradox [3]. It is especially popular for its medicinal properties of anti-inflammation. The functionalization of RSV with different chemical moieties or oligomerization results in stilbenes with myriad properties, suitable for interacting with immune cell components. In monomers, where the hydroxyl groups are located on the B ring is of great importance. With hydroxyl groups in the ortho position, piceatannol (PICE) shows more anti-inflammatory properties than oxyresveratrol (ORV) at the meta position [4]. The hydroxyl groups, however, decrease the ability of derivatives to penetrate cells and for reactive oxygen species (ROS) clearance [5]. Pterostilbene (PTS), on the other hand, has two methoxyl groups at positions C-3 and C-5 on the stilbene skeleton, which provide it with potent anti-oxidative properties due to electron density enrichment [6]. The association between the ring-C/D of amurensin H and Spleen tyrosine kinase (Syk) makes it an ATP-competitive inhibitor, allowing it to inhibit inflammation via NF-κB signaling [7,8]. The addition of a glycoside moiety in the case of polydatin (PD), astringin A, and isorhapontin reduced the anti-inflammatory activity [9]. However, RSV can be toxic at higher concentrations, so it is well worth exploring the rich inventory of stilbenes that plants have to offer for ones that show a balance of properties.

## 2. Diversity of Stilbenes

RSV has two spatial conformations: the most prevalent is *trans*-, which is naturally synthesized in red grapes and peanuts [10]. The *cis*- isomer was found in only a modest percentage of red wines, from the grape cultivars Nebbiolo, Red Globe, and Beauty Seedless [11]. RSV also occurs in the glycosidically bound form in Riesling wines, such as *trans*- and *cis*-3,5,4′-trihydroxystilbene 2-C-glucosides, and the dimers pallidol 3-*O*-glucoside 6a and pallidol-3,3″-*O*-diglucoside [12]. Different functional groups on RSV create diverse stilbenes. The genus *Bletilla* has as many as 164 stilbenes [8]. Much more potent effects are produced by stilbene metabolites than by their parent substances, as these forms are predominant in biological fluids [13]. RSVs most prevalent natural variant is a glycoside form called polydatin (PD) [14], which has strong anti-platelet, antioxidant and anti-inflammatory properties [15]. The RSV glycoside PD was found in berry skins at a 2–4-times higher level than aglycone forms, and it is more soluble during juice extraction [16]. *trans*-Astringin is a stilbenoid that is PICE substituted with a beta-D-glucosyl residue [4]. ORV can undergo hemisynthesis into three glucuronide metabolites: *trans*-ORV-2′-*O*-glucuronide, *trans*-ORV-4′-*O*-glucuronide and *trans*-ORV-3-*O*-glucuronide. Hemisynthesis of gnetol (GN) produces two compounds: *trans*-GN-3-*O*-glucuronide and *trans*-GN-2′-*O*-glucuronide [13]. Oligomerization of RSV also creates stilbenes with potent properties. An 8–8′ coupled RSV dimer from the Vitaceae family is called pallidol [17]. A dehydrodimer stilbene, *trans*-ε-viniferin, is the oxidized RSV formed in woody portions of grapevines under stressful conditions [18]. Viniferins are present dimers and trimers of resveratrol, and their dihydrofuran rings have two stereo chemical centers at positions 7a and 8a [19]. The berries of wild *Vitis* species show greater concentration of *trans*-ε-viniferin [20] than the cultivated *V. vinifera* L. δ-Viniferin is a 3–8′ coupled RSV dehydrodimer [17]. Viniferins have stronger therapeutic and antioxidant activities than RSV, but may be more toxic [21]. Orchidaceae-family plants are known to contain the (dihydro-) stilbene derivatives [22].

Stilbenes can have different groups conjugated to them naturally as well. The stem of *D. usambarensis* F. White has a 3″-methoxycochinchinenene H conjugated chalcone stilbene, along with RSV [23]. The stilbene content in a plant can differ between parts of the plant, stages of growth and season as well. The rhaponticin content found in *R. rhaponticum* L. roots seems to increase from April to October [24,25]. Grape skins have the most stilbenes at maturity due to the increase in the enzymes stilbene synthase, 4-coumarate-CoA ligase and phenylalanine ammonia-lyase [26]. In grapevine, overexpression of the Vitis stilbene synthase1 (Vst1) gene can increase RSV [27]. The formation of phenoxyl radical intermediates through oxidation and their coupling results in oligomerization. Dimers such as ε-viniferin are formed via 8–10′ coupling [19]. In peanuts, RSV content increases on the second day of germination, and additional stilbene isomers are observed [28]. Table 1 summarizes the diversity of stilbenes by plant family, along with the medicinal effects.

## 3. Inflammatory Response

Inflammatory response is a natural biological immune reaction which occurs in our body when it comes into contact with any noxious material. Inflammatory response can be triggered by many factors, such as the entry of pathogens, damage in host cells, exposure to toxic compounds, etc. It is an important response which takes place to maintain the homeostasis of the body. In response to this, there will be decline in the function of the affected tissue, which in turn leads to the pathogenicity of the disease [48]. At the tissue level, inflammation can be characterized by redness, swelling, heat, pain, and loss of tissue function, which is the result of exudation and extravasation. During this inflammatory process, the leukocytes will be programmed to migrate to the site of inflammation, and the process is called transmembrane migration (migration from blood circulatory system to site of inflammation). When leukocytes reach the site, they are activated, and due to chemical and cell signaling, they release cytokines, which are responsible for inflammatory response [49].

## 4. Mechanisms

An immune response mainly consists of four components: (i) inducers, which are responsible for inflammation, (ii) sensors, (iii) mediators, which relay the information from sensors to the site of response, and lastly (iv) the target site.

Inflammation inducers can be broadly classified into two types: responses which are due to damage in the host cell, tissue, etc., called DAMPs (damage-associated molecular patterns), and responses due to the entry of pathogens, the cell wall components of which (PAMPs—pathogen-associated molecular patterns) trigger the immune system. In both inducers, the mechanism of action remains the same, i.e., when a bacterium enters the body, it is detected by receptors present on immune cells such as Toll-like receptors, which are expressed by the tissue-resident macrophages, and these release inflammatory cytokines and prostaglandins [48].

PAMPs are responsible for the activation of the response. In the host cells, the receptors which are responsible for the detection of these patterns are pattern-recognition receptors (PRRs). These are present in both immune cells and non-immune cells. There are different classes of PRRs such as Toll-like receptors (TLRs), C-type lectin receptors (CLRs), retinoic acid-inducible gene (RIG)-I-like receptors (RLRs) and NOD-like receptors (NLRs). Among all of these receptors, TLRs are the most commonly studied receptors in the activation of inflammatory response. When a signal is transmitted from DAMPs and PAMPs, in both cases, transmission is mediated by TLRs, along with myeloid differentiation factor-88 (MyD88). This results in the activation of a series of intracellular signal cascade events, which leads to the migration of transcription factors such as activator protein-1 (AP-1) and NF-κB or interferon regulatory factor 3 (IRF3) into the nucleus, and triggers transcription and inflammation [48].

Once the receptors are triggered and intracellular signaling cascade starts, there are three major pathways which a cell takes to produce inflammatory mediators. These are: mitogen-activated protein kinase (MAPK), nuclear factor kappa-B (NF-κB), and Janus kinase (JAK)-signal transducer and activator of transcription (STAT) pathways [48].

### 4.1. Nuclear Factor Kappa-B (NF-κB) Pathway

The nuclear factor kappa-B (NF-κB) pathway has an important role in cell proliferation, inflammation, immune response and apoptosis. It consists of five different transcription factors: P50, p52, RelA (p65), RelB and c-Rel. Under normal conditions, when there is no trigger for inflammation, the IκB protein present in the cytoplasm is inhibited by NF-κB [50,51]. When the PRRs are triggered, they activate IκB kinase (IKK), which is made-up of two kinase subunits and a regulatory subunit [52]. IKK regulates the NF-κB pathway by means of the phosphorylation of IκB, which results in degradation by the proteasome, leading to the release of NF-κB, which translocates to the nucleus for the transcription of genes. This pathway regulates the production of pro-inflammatory cytokines and inflammatory cell recruitment, which contribute to the inflammatory response [53] (Figure 2).

### 4.2. MAPK Pathway

This involves a family of serine/threonine protein kinases that trigger cellular responses to a variety of stimuli like osmotic stress, mitogens, heat shock and inflammatory cytokines. They regulate cell proliferation, death, survival and apoptosis. MAPKs include extracellular-signal-regulated kinase ERK1/2, p38 MAP Kinase, and c-Jun N-terminal kinases (JNK). Each MAPK signaling pathway consists of three components, namely MAPK, MAPK kinase (MAPKK), and MAPK kinase kinase (MAPKKK). When PRRs are activated, MAPKKK is phosphorylated, which produces phosphorylated MAPKK, which in turn activates MAPKs. Then, the activated MAPKs will finally phosphorylate and activate the p38 transcription factor, which translocates into the nucleus and activates the genes that result in inflammation [53,54] (Figure 2).

### 4.3. JAK-STAT Pathway

This is the pathway through which extracellular factors directly influence gene transcription. When the JAK receptors’ extracellular domain comes in contact with a ligand, there is phosphorylation of the cytoplasmic domain, which opens the active site for the attachment of STAT. Then, STAT undergoes phosphorylation and dimerization for DNA binding capacity, for which tyrosine phosphorylation is essential. After dimerization, STAT will translocate inside the nucleus and bind to DNA, which encodes genes responsible for inflammatory response [53,55] (Figure 2).

Inflammation can occur throughout the body, either systemically or locally. The specific reactions may vary according to the tissue or cell type and the invading pathogen, with common underlying pathways being activated.

## 5. Effect of Stilbenes on Various Types of Inflammation

### 5.1. Inflammation in Macrophages

In inflammation, macrophages have three major functions: antigen presentation, immunomodulation and phagocytosis, by means of cytokine and growth factor production. Macrophages are critical to inflammation’s onset, duration and termination. Deactivation of macrophages is achieved by anti-inflammatory cytokines (transforming growth factor β and interleukin 10) and cytokine antagonists, which macrophages mostly generate [56]. When stilbenes were used to treat macrophage inflammation, the following cell components were especially affected. 

The NF-κB signaling pathway, as covered, is one of the best understood immune-related pathways. It is a critical regulator of autoimmune illnesses as well as of the inflammatory response to infections and malignant cells. It is crucial for innate immune responses in first-responder cells such as macrophages, because the canonical NF-κB response is significantly faster than non-canonical signaling [57,58]. The production of interleukin 6 (IL-6) and cyclooxygenase 2 (COX-2) is mediated by this pathway [57,58].

Sirt1 is a sequential post translational regulator that deacetylates proteins [59]. It promotes the generation of ROS, which causes the aberrant activation of the NLRP3 inflammasome, which will lead to inflammation [60]. NLRP3 recognizes stress and activates caspase-1, which triggers pyroptosis and cytokine release [61]. Lipopolysaccharide (LPS) is an outer-membrane component of Gram-negative bacteria. The proinflammatory portion is a glycolipid composed of a polysaccharide O-antigen, a core oligosaccharide and a highly conserved lipid A moiety. LPS activates cells of the innate immune system, such as macrophages and neutrophils, which synthesize IL-1β, TNFα, MMPs and free radicals that lead to dramatic secondary inflammation in tissues. RSV was used to treat mice which then had the brain infiltrating mononuclear cells ex vivo MOG restimulated. These produced less pro-inflammatory IL-17A and IL-6, which are characteristic cytokines in experimental autoimmune encephalomyelitis (EAE) [62]. In mouse encephalitogenic CD4^+^ cells, SK1 was markedly downregulated. RSV docks to the substrate binding pocket of SK1, directly inhibiting its catalytic activity [63] (Figure 3). This hinders localization to the plasma membrane, where it can access its sphingosine substrate [64]. In rat multiple sclerosis (MS) models, RSV reduced inflammation by SIRT 1 activation, a pathway that increases non-inflammatory cells in brain and peripheral tissue [65]. RSV-induced SIRT1 can also prevent the activation of T cells and NF-κB in colitis brought on by DSS (digestive tract inflammation) [66], as well as deacetylate c-jun transcription factor, which deactivates T cells in a collagen-induced arthritis model [67]. It does this by lowering the SIRT1 Michaelis constant for acetylated substrates [68], such as p53 [69]. RSV, (+)-ε-viniferin and hopeaphenol inhibited the damaging effects of DPPH radicals and prostaglandin E2 in rabbits with arthritis induced by LPS [19,70].

LPS was used in several experiments to induce a state of inflammation. In human THP-1 macrophages induced with LPS, RSV epigenetically regulates survival and apoptosis, and increases the anti-inflammatory IL-4 and IL-10, and miR-Let7a levels. TNFα and IL-6 were reduced [71] (Figure 3). Similar results were seen for *cis*- and *trans*-gnetin H RSV derivatives [72]. *trans*-resveratrol-3-*O*-glucoside is also a Sirt-1 activator, and along with RSV it can inhibit ICAM1 and TNFα induction. This mechanism was used in reducing inflammation in LPS-stimulated macrophages pre-treated with the extract of *Sambucus ebulus* L. fruit [73]. Resveratrol-4′-*O*-glucuronide, however, can upregulate the mRNA levels of macrophage inflammatory protein 1β (MIP-1β) [74]. 

*Macaranga siamensis* S. J. Davies contains many stilbenes unique to its species, among which macasiamenene L shows the best inhibition IC_50_ at a concentration of 1.8 ± 1.1 μmol/L in cell viability studies with the THP-1 cell line [41]. Meanwhile, 1 μmol/L macasiamenene F reduces TNFα from LPS-stimulated macrophages by 20%, and interferes with the DNA binding site of NF-κB. It shows anti-inflammatory properties through modulating IκBα/NF-κB signaling, by degrading IκBα. 3,3′,4,5′-tetramethoxy-*trans*-stilbene (3,3′,4,5′-TMS) and 3,4′,5-trimethoxy-*trans*-stilbene (3,4′,5-TMS) are two methoxy derivatives/analogues of RSV. RAW 264.7 cells stimulated with 1 µg/mL LPS showed enhanced NO release, with upregulated phosphorylation for MAPK pathway members p38, JNK and ERK. A 50 µM pretreatment of the cells for 4 h with 3,3′,4,5′-TMS showed significant suppression for p38 and JNK, and to a lesser extent for ERK as well. 3,4′,5-TMS could suppress expressions of all three proteins [75]. Regarding the NF-κB signaling pathway, pretreatment of 3,3′,4,5′-TMS and 3,4′,5-TMS decreased p-IKKα/β, p-IκBα, p-P65, and MDA accumulation, NF-κB p65 nuclear translocation and intracellular H_2_O_2_. Only 3,3′,4,5′-TMS showed some reversal of SIRT1 expression and decreased the O_2_^•−^ production. RAW 264.7 cells differentiated in response to LPS, taking on an irregular form with pseudopodia and speeding up spreading, which could be decreased by treatment with both compounds [75]. 

t-ORV is able to control the NF-κB signaling pathway to decrease production of NO, TNFα, monocyte chemoattractant protein-1 (MCP-1), IL-1β, inducible nitric oxide synthase (iNOS), CXCL10 and IL-6 in a macrophage cell line and microglial cells induced with LPS [33,58]. t-ORV treatment causes the activation of all three pathways of NF-κB, MAPK and PI3K/AKT/p70S6K signaling pathways to suppress C-X-C motif chemokine ligand 10 (CXCL10), along with other pro-inflammatory cytokines [33]. Macrophages were treated with t-ORV and its glucuronide metabolites t-ORV-4′G, t-ORV-2′G, and t-ORV-3G, all of which could reduce IL-1β levels by 30–50%. However, a 4-fold concentration of glucuronide metabolites was required over the parent compound. The only compounds capable of reducing TNFα were t-ORV and t-ORV-3G. In the same way, the glucuronide metabolites of GN, namely t-GN-2′G and t-GN-3G, significantly reduced IL-1β (32–47%) and TNFα (around 65%) [13]. Moreover, GN has the ability to lower colorectal cancer cell viability and suppress platelet aggregation, platelet-collagen adhesion, and COX-1 activity [34,76].

Hopeaphenol, isohopeaphenol, PICE, and ε-viniferin significantly decreased LPS-induced TNFα and IL-1β production in RAW 264.7 cells, with isohopeaphenol being the most potent against IL-1β [4]. The expression of the TNFα induced by LPS is more significantly suppressed by PICE than RSV [77], and it also inhibits T-cell receptor signaling in primary murine splenocytes [78]. Structurally, the phenyl rings’ hydroxyl groups are important to suppress cytokines production [77]. Other properties of PICE include the inhibition of iNOS expression when bEnd.3 cells (endothelial cells isolated from brain tissue) were pre-incubated with 50 µM of the stilbene for 4 h and stimulated with 1 µg/mL LPS exposure. Only 30 min of pretreatment could reduce p38, JNK, IKKα/β, IκBα and p65 phosphorylation caused by 10 µg/mL LPS. The LPS-triggered nuclear translocation of NF-κB and ROS increase were also prevented [79]. Bavienside A treatment resulted in a significantly reduced amount of NO that LPS-induced RAW264.7 cells produced, with an IC_50_ value of 6.23 μM [80]. Bone marrow-derived macrophages (BMDMs) were stimulated with LPS (1 μg/mL). Here, 10 μM gigantol inhibited NO release by 47%, and decreased levels of TNFα/IL-6 as well, with as little as 1 μM as the minimum effective dose [22]. NO could be inhibited in LPS-stimulated J774.1 cells using the stilbenes found in propolis at less than 0.2% (*v*/*v*). Propolis is an animal product of bees, composed of beeswax and also plant resins, which contribute to its stilbene content. The dihydrostilbene skeletons with catechol moieties in the ethanolic extract of *Senegalese propolis* are the ones that show anti-inflammatory activity. PTS could inhibit the pro-inflammatory responses between RAW 264.7 macrophages and 3T3-L1 adipocytes in vitro [81]. Ampelopsin shows anti-inflammatory effects by inhibiting the NF-κB signaling cascades, phosphoinositide 3-kinase and protein kinase B in RAW264.7 cells [82]. Ampelopsin exerts anti-inflammatory properties through modulating the Toll-like receptor 4 (TLR4) pathway [83] or binding to the ryanodine receptor [84]. 

Stilbenes are also known to attenuate allergies. PICE suppresses the allergic inflammatory response of mast cells through regulating MAPK phosphorylation [85]. Rhapontigenin inhibits histamine release from mast cells, hyaluronidase (HYAL) activity and passive cutaneous anaphylaxis reactions [86]. A derivative of rhapontigenin, called desoxyrhapontigenin, shows anti-inflammatory properties by activating the nuclear factor erythroid 2-related factor 2/heme oxygenase-1 pathway and also in macrophages by attenuating the NF-κB and MAPK pathways [35]. *Agonis flexuosa* (Willd.) leaves contain the stilbenes (*Z*)-2,3-dihydroxystilbene-5-*O*-β-D-glucoside, (*Z*)-pinosylvin mono methyl ether and (*Z*)-pinosylvin-3-*O*-b-D-glucoside. In an in silico study conducted on the relationship between stilbenes and cellular receptors, they showed free binding energies between −11 and −31 kcal/mol when docked with human histamine H1 receptor as histamine blockers for treating allergies and inflammatory conditions. The first glycoside was also able to inhibit in vitro histamine production with an IC_50_ of 0.16 mM in U937 human monocytes [87].

### 5.2. Inflammation in the Liver

Liver injury accounts for 2 million deaths per year globally, with half due to cirrhosis alone [88]. The liver can also sense and respond to systemic inflammation [89]. When the complement system is activated by pathogens or antibodies, it unleashes a cascade of processes that result in inflammation. The liver is in charge of generating the majority of the complement system proteins. Specifically, C9 of the terminal complement complex C5b-9 is a known pro-inflammatory trigger [90]. The NF-κB pathway regulates COX-2 production and maintains antioxidant defenses by controlling ROS-scavenging proteins [91]. It shows antiapoptotic activity by controlling the c-Jun N-terminal kinase (JNK) cascade, thereby suppressing TNFα [92].

To model inflammation in the liver, different experiments have been conducted. CCl_4_ treatment is used to induce inflammation and it has myriad effects in hepatocytes. It causes upregulation of the MAPK/JNK pathway [93], JNK/cPLA2/12-LOX inflammatory pathway [94], and Akt/NF-κB pathway [95], phosphorylation of ERK1/2 and Smad [6], and mRNA expression of C3 and TCC components, platelet-, and leukocyte-type 12-LOX [94]. Normally, arachidonic acid (AA) that has been esterified appears in membrane phospholipids, which, when disturbed by external stresses, becomes de-esterified and liberated by cytosolic phospholipase A2 (cPLA2). Free AA is transformed into pro-inflammatory eicosanoids such as leukotrienes (LTs) and hydroxyeicosatetraenoic acids (HETEs) by the enzymes lipoxygenases (LOX), cyclooxygenase (COX) and cytochrome P450 enzymes (CYP450). By activating neutrophils and macrophages, these eicosanoids exacerbate liver inflammation [94] (Figure 4).

CCl_4_-treated Sprague Dawley (SD) rats’ liver tissue was treated with a 300 mg/kg BW dose of PTS. This reduced the phosphorylation of ERK1/2 and Smad, and the formation of TGF-β, p-ERK1/ERK1, p-ERK2/ERK2, p-Smad1/Smad1, and p-Smad2/Smad2 proteins in the liver 2.64-, 4.36-, 2.10-, 7.47-, and 5.83-fold, respectively [6].

The signaling pathways MAPK/JNK pathway [93] and Akt/NF-κB pathway [95] are responsible for AA metabolism, which is then followed by hepatotoxicity, neutrophils infiltration and M1 polarization of macrophages. This is countered by gigantol, which also reduces the mRNA expression of C3 and TCC components, and platelet and leukocyte-type 12-LOX, responsible for oxidative and endoplasmic reticulum-mediated inflammation. By inhibiting the JNK/cPLA2/12-LOX inflammatory pathway, gigantol reduces the liver damage caused by CCl_4_ exposure [94]. Gigantol keeps C9 in check, and thus prevents the deposition of C5b-9 around hepatic vessels and promotes the formation of CD59b, which protects cells from complement attack in response to inflammation [22].

Subacute liver failure (SALF) is characterized by ascites and the poor regeneration of hepatocytes. As a SALF model, Wistar rats were first injected with thioacetamide (TAA), and then a mixture of 5 mg/kg *trans*-RSV and *trans*-ε-viniferin was supplemented in 3 doses. TNFα, COX-2 and iNOS were reduced and the transcription of the anti-inflammatories IL-10 and NF-κB was upregulated [91]. Steatosis is lipid buildup in the liver, leading to inflammation of the organ. This is characterized by the increase in blood alanine aminotransferase (ALT) and aspartate amino transaminase (AST) levels. Adipose tissue fat can also generate inflammation mediators such as TNF-α, IL-6 and leptin, which can trigger liver injury [96]. ε-Viniferin, found in *Vitis vinifera* L. grapevines, is known to reduce all of these enzymes. To model steatosis, C57BL/6 mice were given a methionine- and choline-deficient diet, leading to lipid accumulation and then inflammation. Meanwhile, 35 and 70 mg/kg *trans*-2,3,5,4′-tetrahydroxystilbene-2-*O*-glucoside (TSG) can reduce the amounts of triglyceride, cholesterol and free fatty acids, AST and ASC, an adaptor molecule. Its inhibitory effect on IL-1β can be seen at 17.5 mg/kg and on IL-18 only at 70 mg/kg [97]. TSG also shows its medicinal effect on acetaminophen-induced liver injury induced in C57BL/6 mouse liver [98]. Pretreatment with 60–180 mg/kg TSG reduced liver injury, cell degeneration, and necrosis by reducing the production of cytokines such as IL-10, IL-6, CCL3, IL-12, IFN-γ, CCL11, IL-2, IL-3, IL-17, IL-1β, GROα/KC, and TNFα. Inflammation was also induced in mice using bicyclol. Meanwhile 40 mg/kg gigantol showed improvement in the infiltration of inflammatory cells, tumefaction, and centrilobular necrosis. The inhibitory effect of gigantol was seen against raised levels of the expression of cytokines’ mRNA (TNFα, IL-1β and IL-6) and chemokines (ICAM-1 and MCP-1) due to bicyclol [22]. 

ORV, RSV and MulA were used in the treatment of LPS-stimulated mice. Here, 80 mg/kg stilbenes which were intraperitoneally injected reduced MDA, ALT and AST levels. MulA showed a particularly suppressive effect on serum transaminase. Acute liver injury (ALI) comes about due to the NF-κB signaling pathway’s activity and LPS/D-GalN. MulA, followed by ORV and RSV, restrained TLR4 and MyD88, which halts the entire cascade, along with reducing amounts of IL-6 and IL-1β. Regarding the MAPK signal pathway, all three of the stilbenes could prevent the phosphorylation of p38, ERK1/2, and JNK, thereby deactivating the downstream pathway [99] (Figure 5).

### 5.3. Inflammation in the Cardiac Tissue

H9c2 cells derived from rat heart tissue were treated with 0.4 mM of palmitic acid, which increased the levels of IL-6, TNFα, IL-1β, Smad3 phosphorylation, and nuclear NF-κB/p65. p65, a subunit of NF-κB, binds to DNA to enhance inflammatory cytokine expression [100]. TSG is able to reverse this effect by reducing NF-κB/p65 levels and downregulating p-Smad3/Smad3 [101]. RSV and PICE, at concentrations ranging from 80 nM to 7 mM, were able to improve cell viability by 95% in H9c2 cells kept in hypoxic conditions for 48 h [102]. Meanwhile, 10–20 μM of PICE is able to inhibit the effects of COX 2 and PI3K signaling in human aortic smooth muscle cells (HASMCs) [103]. It selectively inhibits Syk, which regulates inflammation in hematopoietic cells [104]. It can also regulate iNOS and AP-1 and inhibit IL-1β, IL-18, NO, IL-8, Il-6, TNFα and PGE2. Reduction in cytokine level was observed in LPS-treated peripheral blood mononuclear cells (PBMCs) due to the action of 100 μM of 3″-methoxycochinchinenene H, a *D. usambarensis* F. White stem stilbene. It was found to be more potent than standard drugs [23]. The gut microbiota has a direct effect on coronary heart diseases as it generates trimethylamine (TMA), which is reduced to trimethylamine N-oxide (TMAO), and increasing concentration of which can increase the risk of atherosclerosis. The administration of stilbenes can reduce trimethylamine N-oxide by regulating the TMA-producing microbes in the gut, with resveratroloside, rhaponticin and TSG showing the best ability [105].

### 5.4. Inflammation in the Connective Tissue

Chondrocytes are specialized cells found in certain cartilage tissues such as the intervertebral discs. They are surrounded by collagenous fibers, and they produce substances which make the cartilage strong and flexible. In mouse models, the induction of inflammation causes chondrocyte activity to be hindered, resulting in a reduction in bone mass. The effects of stilbenes on inflammation are listed in Table 2.

### 5.5. Inflammation in the Nephrons

The inflammatory responses seen in the kidney are usually seen as a result of hyperglycemia, causing apoptosis of podocytes due to increased ROS [109]. This induces TNFα production, due to which the glomerular permeability barrier is harmed [110,111]. Inflammatory genes including cytosolic phospholipase A2 (cPLA2) and COX-2 are increased as a result [112], as well as JNK1/2, NF-κB (p65) and ERK1/2 phosphorylation. COX then controls AA conversion to prostaglandin E2 (PGE2). NO is produced by the enzyme iNOS by using arginine and oxygen, which is activated by endotoxins or cytokines. Peroxynitrite (ONOO^−^, RNS) is created when NO reacts with free radicals (•O^2−^) that can damage the cells. 

Following a 1 h pretreatment with ampelopsin C (AC), ampelopsin F (AF), or PD, rat mesangial cells were incubated with TNFα. This reduced cPLA2 protein formation but not its mRNA levels. PGE2 and the three phosphorylated signaling entities were also reduced. The RSV derivatives ampelopsin F (AF) and ampelopsin C (AC) showed an inhibitory effect on cPLA2/COX-2/PGE2 activity, even better than the parent molecule. Meanwhile, 2 μg/mL AC and AF inhibited COX-2 mRNA levels via a p38 MAPK-independent pathway [113]. PD can also control cPLA2 and COX-2. TNFα, IL-1β and PGE2 production are decreased as a result [114], as well as iNOS proteins [5]. Extracellular matrix accumulation due to high-glucose conditions is alleviated by PD in RMCs through anti-inflammatory mechanisms [115]. Diabetic nephropathy was modeled in cultured mouse podocytes (MPC5) by incubating them in a high concentration of glucose. A 48 h co-treatment with 10 μM of TSG blocked the NLRP3 inflammasome–IL-1β axis, preventing apoptosis. The Nod-like receptor protein 3 (NLRP3) inflammasome oligomerization and caspase-1 activation resulted in the release of IL-18 and IL-1β [116,117]. The amount of IL-1β was reduced when NLRP3 was knocked down. Due to TSG treatment, nephrin protein levels increased. ROS, MDA levels, caspase-3 activation, IL-1β, pro-caspase-1, caspase-1, NLRP3 and ASC were decreased [118]. 

### 5.6. Inflammation in the Intestine

Inflammation in the intestines is generally in the form of inflammatory bowel disorder (IBD), colitis, metabolic syndrome (MetS), etc. MetS is a debilitating systemic condition, and its inflammatory response can destroy the gut lining, resulting in insulin resistance, which hampers metabolism and hormone production [119]. Associated with the brush border cells is an abundantly available protein, aminopeptidase, or CD13, a specific cytokine receptor on the cell surface. These cytokines are members of the signaling cascade that ultimately results in inflammation. Aminopeptidase mediates the inflammatory response through G-protein-coupled receptors. By cleaving the N-terminals of cytokines, it controls their activity. Moreover, by trimming the peptides attached to MHC class II, it participates in antigen processing and the proliferation and effector function of immune-related cells [44]. Inflammation is also induced in obese conditions, as adipocytokines are produced with the increased number of adipocytes.

The effects of stilbenes on these disorders are studied using mice models and cell cultures in which inflammation has been induced. Administered RSV and PTS at 5% concentrations showed downregulation of aminopeptidase, IL-1β and TNFα on duodenal brush border membrane proteins. Only PTS could downregulate IL-6 [44]. To model obesity, the diet of C57BL/6J mice was rich in fat. The usage of 2.5% DSS induced colitis (Figure 6). The effects observed were fibroblast cell infiltration, necrotizing colitis, loss of weight and crypts, diarrhoea, and bloody stools [120]. This led to the colon shortening with an increased weight-to-length ratio [121]. PTS could prevent colitis by controlling the inflammatory response, fibrosis, and gut barrier functioning. Treatment of dietary PTS (0.005 and 0.025%) reversed these symptoms, decreased the colon weight-to-length ratio, the total number of aberrant crypt foci and aberrant crypts per colon length. It also suppressed thickening of the intestinal wall and colonic edema, and reduced the disease activity index. PTS also reduced the levels of IL-6, TNFα, IL-1β, COX-2, MMP2, TGF-β1 and p-Smad2 in the colonic mucosa, the last two of which are especially involved in fibrosis. TGF-1/Smad signaling allows fibrogenic mesenchymal cells to make more collagen after stimulation by TGF-β1 [120].

PTS decreased TNFα, IFN-γ and IL-1β in diabetic mice models induced by streptozotocin [122]. Other effects of PTS seen in the intestine are the reduction in C/EBP homologous proteins (CHOPs), which are markers of ER stress, collagen deposition and inhibition of the loss of the E-cadherin [106]. PTS is also able to retain the cells coated with Muc2, a significant mucin that goblet cells make to protect the intestinal epithelium [120] (Figure 6). The stilbene glycoside resvebassianol A has a unique sugar unit 4-*O*-methyl-d-glucopyranose, which was obtained from *Beauveria bassiana* (Bals.-Criv.) Vuill (entomopathogenic fungus) which mediates the biotransformation of RSV. When human intestinal epithelial cells and HIEC-6 cells are stimulated with TNFα/IFN-γ, resvebassianol A inhibits IL-1β and IL-6 production. It does not show harmful effects when tested on HIEC-6 and HaCaT cells. RSV’s anti-inflammatory activity is higher at 25 μM, but resvebassianol A is safe for use even at high concentrations [123]. In Caco2 cells, derived from a colon carcinoma, RSV polymers in the 5–25 M range showed suppression of NF-ΚB. The inhibitory power was most seen in δ-viniferin, followed by trimethoxy-resveratrol, ε-viniferin, pterostilbene-transdihydrodimer and then RSV [124]. 

To model metabolic syndrome, Wistar rats were fed a rich fructose diet (Figure 6). They were treated with water-alcohol extraction of the *V. vinifera* L., which has a total stilbene content of 1.519 g/L, including *trans*-RSV and ε-viniferin. There was a difference in the effects of stilbenes seen with early (14th week onwards) and late (19th week onwards) treatment. Abdominal fat showed a reduction of 53.6% and 40% from the 14th and 19th weeks, respectively. Similarly, TLR4 concentration was reduced by 65.1% and 57.9%, respectively. Systemic inflammatory reaction markers, namely C-reactive proteins, which are useful prognostic indicators for cancers, showed levels that were 76.0% and 81.61% lower, respectively [125]. LPS was also used to model intestinal injury. There was a spike seen in the levels of TLR4 and NO. The former was suppressed by rhein at a concentration of 100 mg/kg body weight [126]. 

### 5.7. Inflammation in the Lungs

Pneumonitis is a non-infectious inflammation of the lungs. Smoking can cause buildup of cholesterol intracellularly, which can lead to the inflammation of the lungs in chronic obstructive pulmonary disease (COPD) [127]. The total leukocyte count was increased in the lungs when mice were exposed to LPS and a 250–300 ppm density of smoke daily for 1 h for 26 days. With the application of amurensin (5–20 mg/kg) or RSV (10 mg/kg) an hour before smoke exposure, a decrease in IL-6, IL-17A, IL-1β, TNFα, IFN-γ and the IFN-γ/IL-4 ratio, a restoration of Th1 bias and improved airway inflammation were observed. This was due to the stilbenes’ inhibition of p-Syk and its inflammatory factors NF-κB p65, NF-κB, and p-NF-κB [7]. Amurensin H is an anti-autophagy agent that regulates Sirt1 and FoxO3 levels, suppresses oxidative stress in CS-induced autophagy models and prevents COPD progression [128]. Upon administration of 5 mg/kg RSV encapsulated within a lipid-core nanocapsule post LPS exposure, the rise in pulmonary elastance was inhibited and there was a reduced concentration of leukocytes in the lungs. Pretreatment with RSV improved lung function, with a decrease in MDA levels and inflammatory cell infiltration, IL-6, TNFα, MCP-1, MIP-2, RANTES, macrophage inflammatory protein-1 alpha (MIP-1α) and KC chemokines. There was a restoration of the catalase level in lung tissue by reducing the phosphorylation of Akt, ERK and p38 [129]. These experiments show that when stilbenes are administered to living systems such as the lungs before exposure to compounds or factors which trigger inflammation, the sudden upshoot of the inflammation factors and accumulation of leukocytes can be reduced.

### 5.8. Inflammation in the Nervous Tissue

Neuroinflammation is an immune response seen in the spinal cord and brain, which results from the release of inflammatory factors such as ROS, cytokines, chemokines and secondary messengers, which carry messages for inflammation. These factors are mostly produced by the CNS, the glial or microglia cells and the CNS’s innate immune cells. Neurotrophins are the growth factors which are produced in vertebrate brains, and they regulate the proliferation of neural progenitors and cell death. Antioxidants which are produced by the mitochondria, along with neurotrophins, have a neuronal regenerative activity and regulate apoptosis [130]. The inflammatory cytokines produced by astrocytes and activated microglia are responsible for neuroinflammation, which results in degeneration of the blood–brain barrier, which leads to the recruitment of leukocytes and other immune cells, leading to inflammation within the brain and spinal cord.

RSV has shown positive results as an anti-neuroinflammatory agent. Its administration causes low regulation of peroxisome alpha-1 (PGC1α) and hypoxia induction factor 1 (HIF-1). These are key modulators of microglial and inflammation in the CNS. HIF-1 increases IL17 [131] and PGC1α downregulates mitochondrial antioxidant genes, resulting in increased oxidative stress and the activation of the NF-κB pathway and NK receptor (p58), making the cells more sensitive to proinflammatory cytokines [10]. RSV also decreases positive iba-1 cells, IL-6, IL-12 and IL-23 required for the dendritic cells and microglia to improve their potential as APC, which can activate T cells towards an inflammatory response. EAE studies emphasize that resveratrol can modulate or attenuate T-cell response by altering/downregulating the CD28/CTLA-4 and CD80 costimulatory pathways [132]. RSV causes T-cell apoptosis, mediated by the estrogen receptor and aryl hydrocarbon [133,134]. In a spinal cord injury model in rats, RSV promotes neuroprotection against lipid peroxidation mediated by radicals [135]. LPS activation of microglial cells produces cytokines, which results in their proliferation. RSV promotes apoptosis and drives the anti-inflammatory M2 microglial phenotype [136]. It also promotes tolerogenic dendritic cell differentiation, which has an immuno-suppressive effect [137]. RSV also inhibited LPS-induced IL-1β, TNFα, C reactive protein, IL-6, NO and MCP-1 in primary mouse astrocytes, demonstrating its highly anti-inflammatory effect [138] (Figure 7).

BV-2 microglial cells pretreated with 10 μM each of PTS, RSV, acetyl-*trans*-resveratrol (ARES), ORV and TSG were subjected to LPS exposure to induce inflammation. ORV was found to exhibit stronger antioxidant activity compared to RSV due to the presence of an extra hydroxyl group. ARES, however, showed less antioxidant activity compared to RSV due to the loss of three free hydroxyl groups. Macrophage polarization to M1 is a crucial process in the pathological process mediated by the NF-κB and JAK/STATs signaling pathways, which can be downregulated by RSV [139]. These M1 macrophages are proinflammatory cells, as they secrete iNOS and TNF-α. ORV is also able to prevent amyloid β25−35-induced rat cortical neuron damage by decreasing the cytosolic Ca^2+^ levels, glutamate and ROS [140] (Figure 7). Ampelopsin can inhibit LPS-induced neuroinflammation through NF-κB suppression and JAK2/STAT3 signaling in microglial cells [141]. In the LPS-stimulated macrophage and microglial cell model, ε-viniferin, a dehydrodimer of resveratrol, proved to be an excellent therapeutic agent for Alzheimer’s disease (AD), as it can disaggregate amyloid deposits, leading to inflammation [142]. Furthermore, it can decrease NO production by inducing nitric oxide synthase, which is a hallmark of inflammation in tissue [143]. Nitrite buildup is a sign that NO synthase activity is present. BV2 cells were cultured with PC 12 cells induced by LPS, and it was reported that the expression of inflammation mediators such as TLR4, MyD88, and NF-κB was significantly high [144]. BV2, when treated with the stilbene isolated from *Bletilla striata* (Thunb.) Rchb.f., can significantly reduce the NO production and thus the cytotoxicity [144]. Stilbenoid macasiamenene was isolated from *M. siamensis* S. J. Davies and its anti-inflammatory effect on monocyte and microglial cells induced by LPS was studied. These studies confirmed that it can interfere with IκB/NF-κB and MAPKs/AP-1 cascade reactions towards inflammation. Pre-, co- and post-treatment of BV2 microglial cells significantly reduced the secretion of the cytokines TNFα and IL-1β, which was monitored by their gene transcription and translation processes [41].

In stroke, there is a pathogenic function of inflammation due to neuronal ischemia and reperfusion damage (IRI) [145]. PTS can attenuate the effects of inflammation by means of the suppression of Nox2-related oxidative stress and activates the NLRP3 inflammasome [146]. In cases of cerebral ischemia-reperfusion after stroke, inflammation is mediated by astrocytes, which triggers nuclear factor (NF)-κB to express and secrete proinflammatory factors. Hippocampal neuronal HT22/astrocytoma U251 cells and cerebral artery occlusion-reperfusion were treated with PTS to study inflammation. The beneficial effect of PTS was reducing the oxidative stress in cells and subsequently reducing the mediators such as TNFα, IL-1β, and IL-6, while on other hand, PTS can boost antioxidant enzymatic activities such as superoxide dismutase and glutathione peroxidase [147]. Diabetic cognitive impairment is a neurodegenerative disease. PTS can reduce chronic neuroinflammation by suppressing oxidative and carbonyl stress and glial cell activation. It also reduces dopaminergic neuronal loss. It was particularly seen to affect the TLR4/NF-κB signaling pathway [148]. Stroke due to a subarachnoid hemorrhage can injure the olfactory bulb. PICE is seen to suppress cytokines such as TNF-α and IL-6, and components of the inflammatory pathways such as NF-κB and SIRT1, in a rat model of olfactory bulb damage [149]. 

β-secretase, or BACE1, is a protease involved in neuronal function. β-secretase is responsible for the production of toxic β-amyloid (Aβ), which has an impact on Alzheimer’s disease etiology [150]. RSV oligomers obtained from the *Paeonia suffruticosa* Andrews seed coat extract had a DPPH free-radical scavenging activity and could inhibit β-secretase. Vitisinol C, scirpusin, and miyabenol C exhibited anti-aggregative activity which reduces Aβ deposition [151]. β-amyloid protein-induced neuroinflammation was prevented by administering RSV, which activated the PI3-K (phosphatidylinositol-3-phosphate kinase) signaling pathway to inhibit apoptosis [152]. PTS is a strong neuromodulator in Alzheimer’s disease, and mediates elevated peroxisome proliferator-activated receptor (PPAR)-α expression [153]. Using mice with bilateral common carotid artery occlusion (BCCAO), PTS reduced the death of hippocampus neurons and the activation of microglia, TLR4 and the downstream cytokines. There was increased TLR4 and Triad3A–TLR4 interaction ubiquitination and degradation, which could be the key anti-inflammatory target [106]. 

Finally, TSG was found to prevent β-amyloid-induced senile plaque deposition [154]. Mice which were orally treated with APP/PS1 + TSG for 2 months showed a significant decrease in Aβ plaque. Here, 120 mg/kg TSG affected gene regulation, where 324 genes were upregulated, including those responsible for the immune system, antigen processes, cytokine response, apoptosis and NF-κB transcription. Meanwhile, 460 genes were downregulated, some of which were responsible for chromosome segregation, cell cycle and CNS myelination (Figure 7). TSG was also found to reduce TNFα, IL-1β, and IL-6 and increase TGFβ, CX3CR1 and iNOS. In BV2 cells, TSG decreased the levels of IL-5, CCL4/MIP1β, IL-10, IL-1β/IL-1F2, IL-18, IL-1β, IL-2, COX-2, GM-CSF, G-CSF, iNOS, CCL5/RANTES, TNFα, IL-3 and IL-4 [155]. 

In C57BL/6 mice, neuroinflammation was activated by myelin oligodendrocyte peptide (MOG35–55), causing the inflammatory cell influx of Th1 and Th17 to the CNS, with inflammation caused by CD4^+^ T helper cells and then followed by demyelination. Here, 100 mg/kg RSV administration for 15 days lowered the level of circulating pro-inflammatory cytokines [156]. Polyphenol-pretreated mouse neurons and astrocytes cells were exposed to Aβ_42_ and IL-1β, which allowed for the disaggregation of Aβ_42_. Here, 1 μM of RSV and *trans* ε-viniferin could reduce Aβ_42_ induced TNFα by 33% and 54%, respectively, and IL-6 by 25% and 40% [157].

## 6. Effect of Stilbenes on AChE and BuChE

A cholinergic enzyme called acetylcholinesterase (AChE) is located in postsynaptic neuromuscular junctions. It hydrolyzes the neurotransmitter acetylcholine (ACh) to terminate signaling between synapses [158]. Ach, other than mediating neural transmission, has a crucial role in the immune response in inflammation. T cells are also able to produce ACh during viral infection in response to IL-21 signaling. This immune-derived ACh also allows T cells to migrate into infected and cancerous tissues [159]. Butyrylcholinesterase (BuChE) is a serine hydrolase related to acetylcholinesterase that typically hydrolyzes butyrylcholine [160]. In a comparison analysis on AChE inhibition between RSV and PTS (monomers) and δ-viniferin, pterostilbene *trans*-dehydrodimer and pallidol (dimers), galantamine was used as the AChE inhibitor and as a positive control. The outcome was that the dimers showed over 50% more inhibition than the monomers [2]. Studies on the inhibitory effects of synthetic 1,2,3-triazolo (thieno) stilbenes on LPS-stimulated PBMCs, TNFα, AChE and BuChE production revealed that the inhibitory effect of the allyl-thienobenzotriazole photoproduct on eqBChE (from equine serum) is twice as strong as the standard of galantamine [161]. The rhizome of *Iris domestica* L. in the third year of cultivation yields PICE as one of the metabolites. In comparison with RSV, it showed a better overall effect, with an IC_50_ against AChE of 218.93 μM and against BuChE of 74 μM. RSV, on the other hand, proved to be mildly active only against BuChE [162]. In cases of LPS promoting NO generation along with AChE, the molecules thunalbene, *cis*-3,3′-dihydroxy-5-methoxystilbene, and batatasin III of *Pholidota cantonensis* Rolfe were inhibitory [43]. A hybrid compound furocoumarin-stilbene was prepared, and its activity, examined against acetylcholinesterase (AChE) and butyrylcholinestarase (BChE), was reported to be better than that of the individual molecules [163]. PSCE (*Paeonia suffruticosa* Andr. seed coat extract) and its constituent compounds showed a concentration-dependent inhibitory activity against AChE and BuChE, with high selectivity towards AChE. The RSV oligomers had strong inhibition of AChE, in contrast to the parent molecule itself. As a potent tyrosinase, GN also acts as an AChE inhibitor [164,165]. Table 3 summarizes the effects of stilbenes seen in myriad cells and tissues for the suppression of inflammation.

## 7. Bioavailability

Stilbenes, as phenolic compounds, rapidly undergo metabolism, such as oxidation, reduction, and conjugation reactions, in various organs, predominantly the liver. This is the main reason for their poor availability. This property of stilbenes limits their application as therapeutic agents. Although stilbenes showed excellent anti-inflammatory properties, their use in medication is limited due to their poor absorption by the intestines and easy metabolism. For instance, 70% of orally ingested RSV is absorbed, but only 5 ng/mL is finally available, having a 9 h half-life in plasma. It is quickly converted into sulfo- or glucurono-conjugates [172] via extensive hepatic metabolism. Similarly, the total bioavailability of ORV is just 9.1–15.2% [173]. The Lipinski rule of five for drug-likeness demonstrates the similarity of compounds to oral drugs, based on molecular weight, number of hydrogen bond acceptors and donors, and log P. RSV and PICE, also seen in the extract of *Cissus quadrangularis* L., pass this rule [174]. The microflora in the colon metabolizes them, resulting in trans-formed compounds [175] which are further metabolized or excreted by the liver [176]. Phase 2 metabolism creates water-soluble inactive compounds that are easily excretable. These processes are carried out by the enzymes uridine 5′-diphosphate glucuronide transferase, sulfotransferase, and catechol *O*-methyltransferase [13]. The extra hydroxyl group in ORV makes it more susceptible to the glucuronidation and sulfation reactions of this phase than RSV, giving it a poorer pharmacokinetic profile [64]. When the hydroxyl groups of RSV are substituted, the stilbene properties improve. One of the derivatives of stilbene, namely pinostilbene, exhibited better bioavailability with oral administration of 50 mg/kg, which was attributed to its lipid solubility and chemical structure, with only one para-hydroxyl group on cycle B, and the dimethoxy on cycle A provides stability of resistance as well as metabolism [177]. PTS is more easily absorbed than RSV, with four times the bioavailability (80%) [178,179]. Its 3,5 meta-position methoxy substituents increase its lipophilicity. The half-life of PTS is 104 min. Phase 2 enzymes convert PTS to pterostilbene-4′-*O*-glucuronide. Pterostilbene-4′-*O*-sulfate is another form that can be eliminated from the body, and the sulphation of PTS is carried out by human sulfotransferase (SULT). PTS has only one hydroxyl group available for sulfation, compared to RSV, increasing its stability and bioavailability [180]. In mice that have been fed a PTS-containing diet for 3 weeks, it was observed that pinostilbene is the major metabolite left in the colon, attributed to demethylation by gut microbiota [181].

ARES is also more bioavailable than RSV, as acetyloxy groups allow for better penetration into the cell membrane. The glucosidic stilbene TSG is less inhibitory than RSV, as the highly hydrophilic moieties affect cell entry [139]. However, it has better stability and water solubility than RSV [97]. Additionally, an increase in oligomerization generally decreases bioavailability [182]. A far more promising route of delivery is intraperitoneally, as bioavailability was improved to over 90% for 2.5 mg/kg ε-viniferin, which is rich in carbons and hydrogens. This is in stark comparison to a dose of 40 mg/kg orally, which yielded a bioavailability of 0.771% [183]. Encapsulating ε-viniferin in phospholipid-based multi-lamellar liposomes (MLLs) called spherulites or onions lessens the isomerizing effect of UV light on it and increases its water solubility [19,184]. Another way to improve bioavailability is to incorporate stilbenes into polymeric lipid-core nanocapsules. Lung injury model mice treated orally with free RSV-LNC exhibit better bioavailability and improved response to pulmonary infection, while further analysis reported the detection of RSV in lung tissue [129].

## 8. Conclusions

The wealth of natural compounds found in the plant kingdom proves that there is an abundance of resources from which drug discovery efforts can benefit. Stilbenes, along with their derivatives, are proving to be a reliable source of anti-inflammatory agents. The vast majority of plants explored for their curative properties are edible and highly significant to both diet and culture, giving emphasis to the importance of healthy eating habits for health and wellbeing. The mechanistic action on enzymes and cytokines has been laid out through animal models and cell culture studies, showing in detail the effect of the entire plant extract as well as the purified individual compound. Conjugating stilbenes with synthetic molecules can create composites with advanced properties for the highly specific targeting of cellular components. This shows that the path ahead has room for unlimited creativity regarding the composition and delivery systems for stilbene-based drugs in combating inflammation and much more.

## 9. Future Perspectives

Based on the vast knowledge on the diverse modes of action of different stilbenes, they are seen as a viable means to control inflammation. Due to their limitations of bioavailability, further research is required to develop optimized methods of delivery. Synthetic stilbenes were specially designed to assess their effects against the NLRP3 inflammasome and autophagy in BMDMs, J774A.1 cells and the peritoneal inflammation model, triggered by ATP through the AMPK pathway. IIIM-941 inhibits this by preventing the oligomerization of ASC, which is vital for inflammasome assembly and activation. IIIM-983 also shows NLRP3 inflammasome inhibition when cells are primed with LPS. IIIM-941 suppressed monosodium-urate-crystal-induced IL-1β in conditions of localized inflammation without changing the levels of TNFα and IL-6, and also in MSU-induced mouse paw and foot edema as a model of inflammation [60]. Derivatives of stilbenes that were 2,6-dihalogenated with the substitution of pyridine showed over 90% inhibition of inflammatory symptoms [185]. Some stilbene sulfonamides in the concentration range of 0.1 μM to 100 μM showed promise as inhalable drugs in opposition to 2D human lung airway epithelial/immune cells [186]. To combat neuronal inflammation, synthetic composites of a nitroxide spin label for antioxidant activity and a stilbene scaffold for Aβ-O engagement were constructed. They are called paramagnetic amyloid ligands (PALs). When human brain microvascular endothelial cells (HBMECs) were challenged by TGRL lipolysis products, PALs reduced the inflammation. Specifically, PMT-401 suppressed IL-6, IL-8, ATF3 and E-selectin gene expression, and COX-2 expression may be reduced by PMT-401, PMT-302 and PMT-303 [187]. Further techniques to improve the delivery and bioavailability and activity were reviewed by Navarro-Orcajadaa [188].

## Figures and Tables

**Figure 1 molecules-28-03786-f001:**
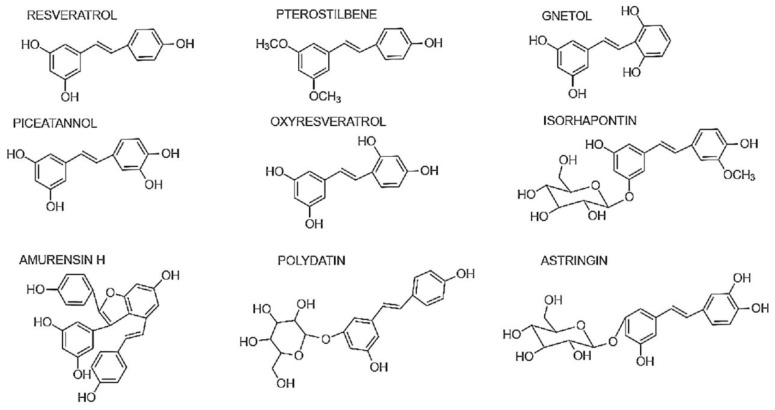
Structure of some important stilbenes.

**Figure 2 molecules-28-03786-f002:**
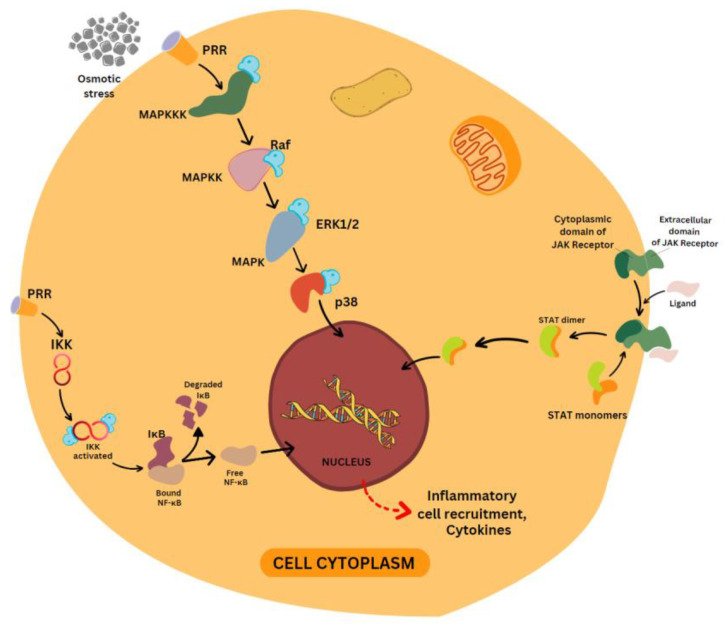
NF-κB, MAPK and JAK-STAT inflammatory pathways. Image created with the help of Canva. NF-κB pathway: pattern-recognition receptors (PRRs), inhibitor of NF-κB kinase (IKK), inhibitor of NF-κB (IκB), nuclear factor kappa-light-chain-enhancer of activated B cells (NF-κB). MAPK pathway: mitogen-activated protein (MAP) kinase kinase kinase (MAPKKK), mitogen-activated protein kinase kinase (MAPKK), extracellular signal-regulated kinase 1/2 (ERK1/2), mitogen-activated protein kinases (MAPKs), Ras effector (Raf). JAK-STAT pathway: Janus tyrosine kinase (JAK), signal transducer and activator of transcription (STAT). The dashed red arrow indicates the final consequence of activation of inflammatory pathway, i.e., the release of cytokines and inflammatory cell recruitment.

**Figure 3 molecules-28-03786-f003:**
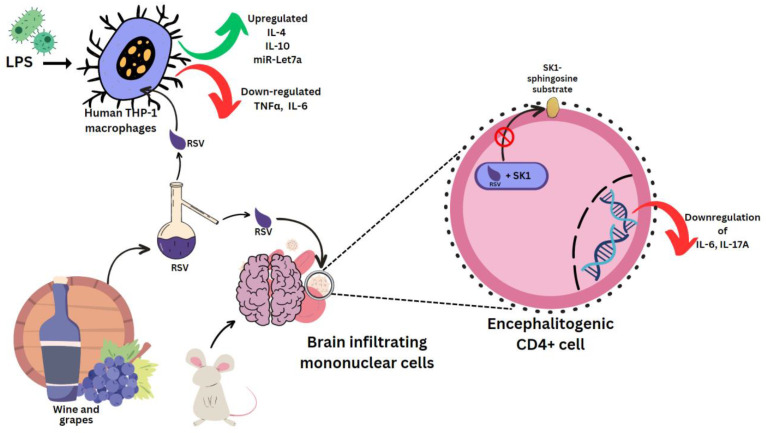
Effect of resveratrol on mouse and human cells. Lipopolysaccharides (LPS), clusters of differentiation 4 (CD4^+^) cells.

**Figure 4 molecules-28-03786-f004:**
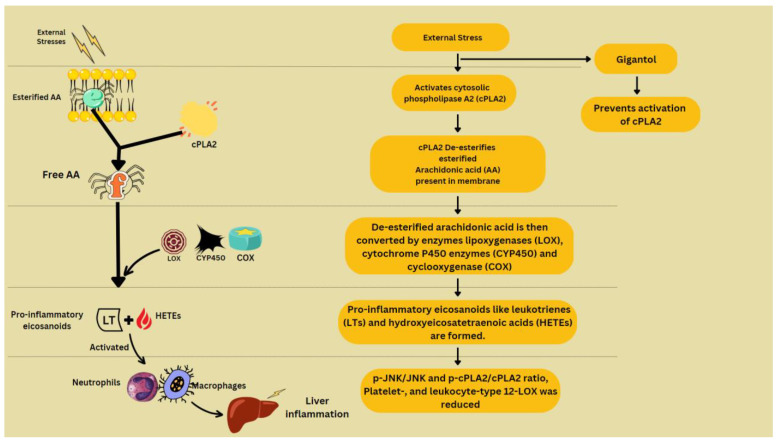
Arachidonic acid metabolism and contribution to inflammation.

**Figure 5 molecules-28-03786-f005:**
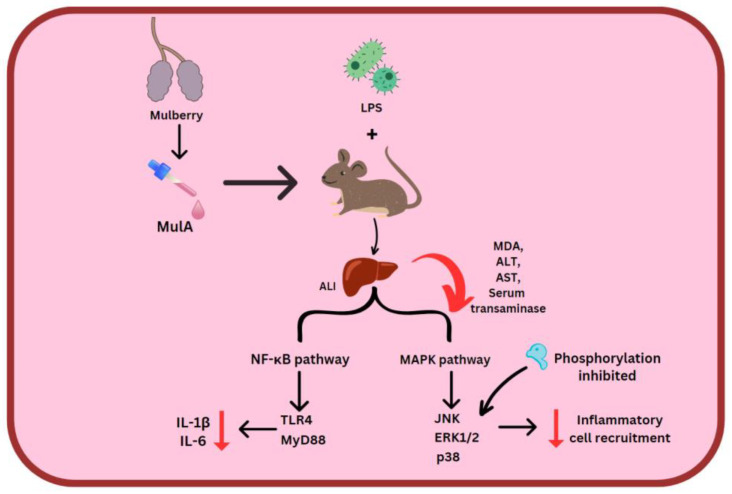
MulA effect on 2 inflammatory pathways. Mulberroside A (MulA), lipopolysaccharides (LPS), acute liver injury (ALI), malondialdehyde (MDA), alanine aminotransferase (ALT) and aspartate amino transaminase (AST), Toll-like receptor 4 (TLR4), myeloid differentiation primary response 88 (MyD88).

**Figure 6 molecules-28-03786-f006:**
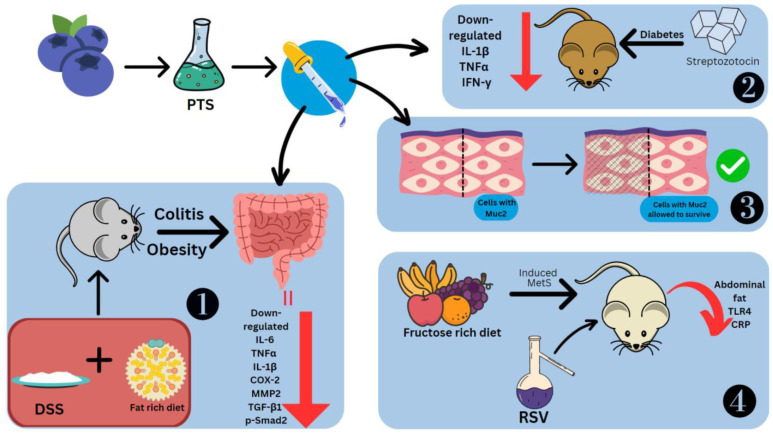
Effect of stilbenes in intestinal inflammation. Pterostilbene (PTS), mucin 2 (Muc2), matrix metalloproteinase-2 (MMP2), phosphorylated mothers against decapentaplegic homolog 2 (p-Smad2), metabolic syndrome (MetS), C-reactive protein (CRP). 1. PTS preventing DSS-induced colitis. 2. PTS decreasing cytokines in Streptozotocin-induced diabetic mice. 3. PTS retaining cells coated in Muc2. 4. RSV reducing Metabolic Syndrome in mice fed on fructose-rich diet.

**Figure 7 molecules-28-03786-f007:**
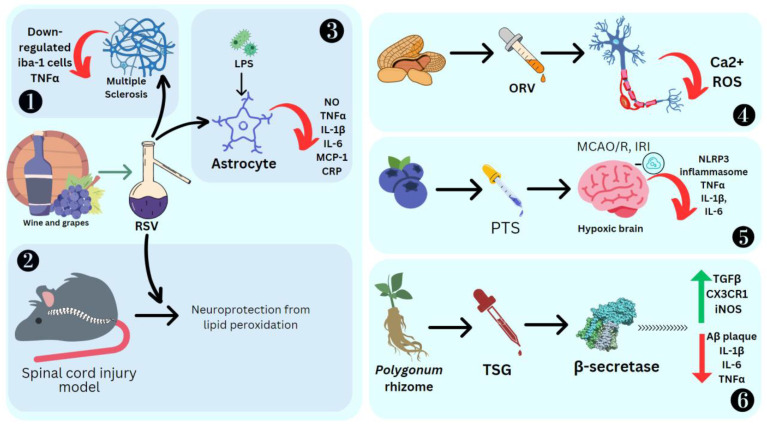
Effect of stilbenes on nervous system inflammation. Ionized calcium binding adaptor molecule 1 (Iba1), oxyresveratrol (ORV), middle cerebral artery occlusion/reperfusion (MCAO/R), ischemia reperfusion injury (IRI), nucleotide-binding domain, leucine-rich–containing family, pyrin domain–containing-3 (NLRP3), *trans*-2,3,5,4′-tetrahydroxystilbene-2-*O*-glucoside (TSG), amyloid-β (Aβ), chemokine (C-X3-C motif) ligand 1 receptor (CX3CR1), reactive oxygen species (ROS). 1. RSV reducing inflammation in Multiple Sclerosis by SIRT1 activation. 2. RSV promoting neuroprotection against lipid peroxidation. 3. RSV reducing cytokines in primary mouse astrocytes stimulated by LPS. 4. ORV preventing amyloid β25−35-induced neuron damage by decreasing Ca^2+^ and ROS levels. 5. PTS reducing MCAO/R induced cytokines. 6. TSG preventing β-amyloid-induced plaque deposition.

**Table 1 molecules-28-03786-t001:** Diversity of stilbenes among plant families with the effect on health.

Family	Plant	Source of Extract	Stilbenes Present	Effects	Ref.
Vitaceae	*Vitis amurensis* Rupr. (Amur grapes)	Leaves, stem and roots	Dimer: Amurensin H	Effective against asthma and chronic obstructive pulmonary disease	[29]
Vitaceae	*Vitis vinifera* L. (European wine grape)	Vine shoot	Monomers: *trans*-PICE Dimers: Ampelopsin A Pallidol *trans*-Scirpusin A Vitisinol C δ-, ω-, ε-Viniferin Trimer: *trans*-Miyabenol C Tetramers: Hopeaphenol Isohopeaphenol, Vitisin A, B	Antioxidant activity, cytoprotective effect against β-amyloid-induced toxicity	[30]
Vitaceae	*Vitis thunbergii* Sieb. & Zucc. (Lobular grape)	Root and stem	RSV, Dimers: Vitisinols A–D (+)-ε-Viniferin Viniferal Trimer: Ampelopsin C Tetramers: Miyabenol A (+)-Vitisin A, (+)-Vitisin C	Anti-platelet, anti-oxidative	[31]
Paeoniaceae	*Paeonia suffruticosa* Andr. (Tree peony)	Seed coat	RSV Dimers: *trans*- and *cis*-ε-Viniferin Trimer: Suffruticosol A, B, C, D *trans*- and *cis*-Gnetin H	Protects against osteoarthritis in chondrocytes	[32]
Moraceae	*Morus alba* L. (Mulberry)	Fruits	Monomers: ORV Mulberroside A (MulA)	Anti-inflammation through MAPK and PI3K/AKT/p70S6K signaling pathways	[33]
Gnetaceae	*Gnetum gnemon* L. (Melinjo)	Roots	Monomer: GN	Reduces cell viability in colorectal cancer cells and inhibits platelet aggregation	[34]
Polygonaceae	*Rheum undulatum* L. (Rhubarb)	Rhizomes	Monomer: Desoxyrhapontigenin	Attenuates the NF-κB and MAPK pathways	[35]
Polygonaceae	*Rheum rhaponticum* L. (Rhapontic rhubarb)	Roots	Monomer: Rhaponticin Desoxyrhaponticin Dimer: Rhapontigenin Desoxyrhapontigenin	Anti-inflammatory, antioxidant and anti-cancer	[36]
Cyperaceae	*Cyperus articulatus* L. (Jointed flatsedge)	Rhizomes and roots	Monomer: PICE, Dimer: *trans*-Scirpusin B Trimer: Cyperusphenol B	Inhibition of NO, iNOS and COX-2	[37]
Fabaceae	*Arachis hypogaea* L. (Tainan 9 peanut cultivar)	First germinated sprout	*trans*-arachidin-1 and -3 IPP IPD Monomer: RSV Dimer: Arahypin-7	Binds with cannabinoid receptors, anti-carcinogenic	[38]
Fabaceae	*Cajanus cajan* L. (Pigeon pea)	Leaves	Longistyline A Cajaninstilbene acid	Prevents MRSA infections	[39,40]
Euphorbiaceae	*Macaranga siamensis* S. J. Davies (Kanda)	Leaves and twigs	Macasiamenenes A, B, F, K, L, and P	Apoptotic effects on THP-1 leukemia cell line	[41]
Poaceae	*Triticum turgidum ssp*. Durum (Senatore Cappelli wheat)	Whole grains	Pinosylvins	Anticancer, anti-inflammatory, antioxidant, neuroprotective, anti-allergic	[42]
Orchidaceae	*Pholidota cantonensis* Rolfe (Rattlesnake orchids)	Whole grass	*cis*-3,3′-dihydroxy-5- methoxystilbene, Thunalbene, Batatasin III	AChE inhibitory, nitric oxide (NO) inhibition	[43]
Caesalpiniaceae	*Chamaecrista pumila* (Lam.) K. Larsen (Dwarf Cassia)	Aerial parts	Chamaecristanols A, B	Laxative, heal wounds, treat ulcers, snake and scorpion bites	[23,44]
Pinaceae	*Picea abies* L. (Norway spruce)	Bark	Astringin Isorhapontigenin Isorhapontin	Inhibits NO production without decreasing oxidative stress	[45]
Ericaceae	*Vaccinium stamineum* L. (Deerberry)	Berry	Pterostilbene Piceatannol	Inhibits activator protein-1 (AP-1) and NF- κB. Induced apoptosis of HL-60 cancer cells	[46,47]

**Table 2 molecules-28-03786-t002:** Effects of stilbenes on connective tissues.

Stilbenes	Concentration	Model System	Observation	Ref.
Amurensin H	4 and 8 μmol/L	IL-1β-induced rat knee chondrocytes	Blocked elevation in IL-6, IL-17, TNFα, TLR4, TRAF6, Syk phosphorylation, NO levels and iNOS expression. Decreased PGE2 and COX-2 levels. Up-regulated COL2A1 and GAG, major components of ECM	[29]
	20 mg/kg	MIA-induced mice	Alleviated bone wear and cartilage loss from 60 ± 13.4% to 25.5 ± 11.4%.	
3,5,4′-Trimethoxy-*trans*-stilbene (BTM)	1% and 0.5%	Rabbit knee joints	Topical administration reduced inflammatory cell infiltration, decreased bone destruction and roughness, and inhibited fibrous connective tissue proliferation. Microemulsion-based hydrogel method decreased IL-1β and TNFα.	[106]
Ampelopsin C	25 μM	Human chondrocytes	Inhibits PGE2 with IC_50_ 15.52 μM	[107]
Desoxyrhapontigenin	50 mg/kg	LPS-stimulated mice and RANKL-induced osteoclastogenesis.	LPS induced trabecular separation, while bone surface and volume changes were attenuated. Anti-osteoporosis activity by inhibiting RANKL. Osteoclast formation is suppressed at an early stage by inhibiting the MAPK/AP-1 signaling pathway, ERK phosphorylation, and the expression of c-Fos and NFATc1.	[108]

**Table 3 molecules-28-03786-t003:** Effect of stilbenes in various animal and cell models of disease.

Stilbenes	Concentration	Model System	Observation	Ref.
ε-viniferin	120 mg/kg	Endotoxin-induced ALI in mice	During lung injury, they decrease inflammation through SIRT3 expression.	[166]
RSV	100 mg/mL	Human umbilical vein endothelial cells (HUVECs)	Reduced amounts of factor VIII, t-PA-1, neutrophils recruiting factor von Willebrand factor (VWF) and IL-8.	[167]
Longistylin A	10 µg	Mice model	Topical application on skin prevented MRSA infection in wounds, decreased IL-6 and TNFα.	[39]
Cannabidiol, luteolin and PICE	Mixture in ratio 10:25:25 µM	Canine epidermal keratinocytes (CPEK)	8 h exposure increased cell viability, decreased DNA methylation in the promoter of *ccl17*, and increased it in *tslp* during inflammation.	[168]
	10:50:50 µM and 10:100:100 µM	Canine Macrophage DH82 Cell Line	Cell viability decreased, methylation increased in one *il31ra* and three *ccl17* sites, and decreased in one *ccl17* site.	
RSV	50 μM	Mice Periodontitis model	Decrease in IL-6 localization pattern, TNFα, MPO, Ki67-positive (tumor tendency), RUNX2-positive and osteoclastic cells. Increases bone tissue mass and healing, osteoblasts, collagen, CD31 cells. Macrophage marker F4/80.	[169]
RSV	30 µM	Cells of vascular smooth muscle stimulated by angiotensin II	Resists hypertrophy via PDGFR, NADPH oxidase, EGFR and c-Src activation and it also links to the ERK1/2 and Akt signaling pathways.	[170]
RSV with NDAT	10 μM of RSV and 10^−7^ M of NDAT	Mouth cancer SCC-25 and OEC-M1 cells	Reduction in IL-1β and TNFα, STAT3 phosphorylation and accumulation. Moderates integrin ligands, which control the immunomodulating PD-1/PD-L1 checkpoint.	[171]
Rhapontigenin	1 µg/mL	HUVECs	Inhibition of 5-LOX	[105]

## Data Availability

Not applicable.

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
