# Peer review of "Stilbenes, a Versatile Class of Natural Metabolites for Inflammation—An Overview"

_molecules, 2023, doi:10.3390/molecules28093786_

Round 1
Reviewer 1 Report
Overall, the manuscript titled “Stilbenes a Versatile Natural Metabolite for Inflammation: An Overview” embodies a good piece of information about stilbenes, and can be accepted for publication with major modification, based on the following points.
· Title: Can be modified as “Stilbenes a Versatile class of Natural Metabolites for Inflammation: An Overview
· Abstract: The length of abstract can be shortened.
· Introduction: Kindly redraw the image given in figure 1, using any software like ChemDraw. This because the 4-OH group in benzene ring of polydatin is added manually not in chemdraw.
· Please follow the journal guidelines and cite the references as per “Molecules” journal format.
· Table 1: Kindly cite reference as per the journal guidelines. It will be better if structure of molecules can be incorporated.
· As it’s a molecules journal, so, a section over chemistry/Characterization of stilbenes can be incorporated: for example, you may refer the work of Shivkanya et al. 2022: https://www.mdpi.com/1420-3049/27/16/5072; Baderschneider et al. 2000: https://doi.org/10.1021/jf991348k; Silvia et al. 2022: https://doi.org/10.1080/10408398.2022.2045558
· Please check the headings or subheadings format for similarity in entire manuscript. Colons are not required
· References: Some more latest references of 2023 can be cited. Kindly cite reference as per the journal guidelines
Author Response
Dear Reviewer
We thank the reviewer/s for evaluating the quality of the manuscript and providing their valuable suggestions for improving the quality of the manuscript.
The Response to the reviewer comment is attached.
Thanks and Regards

Reviewer 2 Report
This is a reasonably well-written review, however, there are some problems that must be corrected. It is not a critical review, but more encyclopedic.
I am not an expert on inflammation, nor will be most readers of this review in the journal Molecules, as it is not a medical journal. Therefore, the use of acronyms and abbreviations without defining them (especially in the abstract) will require many of the readers to look them up.
Some suggestions:
The figure legends give the software with which there are drawn, which is irrelevant. But, many of the figure legends are entirely inadequate.
For example, Fig. 2 should define the abbreviations and tell the reader what a dashed arrow means.
Fig. 3. – tell the reader what 2 and 4 are. Where are 1 and 3?
There are similar problems with most of the other figures.
Throughout the paper, the authors capitalize the spelling of many, but not all, compounds (e.g., amurensin H). The names of compounds should not be capitalized.
Line 55 – change to……are examining natural compounds, especially….
Line 60 - Phytoalexins are low molecular weight antimicrobial compounds that are produced by plants as a response to biotic and abiotic stresses. Phytoalexins are not made to protect against radiation, injury, or stress. This does not mean the radiation, injury, and general stress might not be a selection pressure for evolution of production of some stilbenes.
Lines 66-67 – Reword this sentence.
Lines 84-85 - …found in only…
Table 1 – Add Vaccinium species, as they are well-known as a major source of pterostilbene in the diet.
Line 119 – effects
Line 126 – At the tissue…
Lines 672-673 – Is there really good proof that degradation is the main reason for poor bioavailability. The main reason for low bioavailability of resveratrol compared to pterostilbene is relatively low lipophilicity. How do the various stilbenes score by Lipinski’s rule of five for pharmaceuticals?
Line 694 – What is PTS degraded to? The most obvious degradation product would be resveratrol, as animals have demethylases. If so, demethylation in some cellular compartments might be beneficial to biological activity.
Lines 707-708 – This is a naïve and unscientific statement.
Line 713 – Be critical. Most claims of synergism are not rigorous.
714-716 – This anthropomorphic claim is specious.
Section 9 – This should be incorporated into earlier text.
The formats of references are inconsistent; e.g., compare ref. 1 to 8.
Author Response

(The authors gave the same response as above.)

Round 2
Reviewer 1 Report
Revised manuscript can be accepted for publication.
Reviewer 2 Report
The authors's responses to my suggestions and criticisms are adequate.